# Rapid and Accurate PPA Prediction for the Template-Based Processor Design Methods

**Mingxin Tang** †📵**, Libo Huang and Wei Chen** *,†

College of Computer Science and Technology, National University of Defense Technology, Changsha 410073, China
* Correspondence: chenwei@nudt.edu.cn
† These authors contributed equally to this work.

**Abstract:** The template-based chip design method aims to build rapidly. However, it still need synthesis and simulation flows to get the performance, power, and area (PPA) reports and find the proper parameters set in the design space exploration, which takes a long time. Therefore, a rapid and accurate PPA prediction method is proposed. At first, the PPA Prediction Model based on Multivariate Linear regression (ML-PM) is proposed to fit the multiple parameters' influence on the PPA via the single parameter affection. Moreover, a Multivariate NonLinear regression Prediction Model (MNL-PM) based on Amdahl's law is introduced to improve the accuracy of the PPA estimation. The empirical evaluation of the method shows that the PPA prediction for the template-based chip design methods can reach 98.60%, 99.19%, and 98.53% accuracy on performance, power, and, area separately, when compared with the PPA generated via the synthesis and simulation flows.

**Keywords:** PPA prediction; template-based methods; multivariate regression; agile chip design

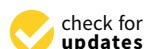



## 1. Introduction

Currently, the Dennard scaling law [1] and Moore's law [2] are gradually coming to an end, and the agile development of hardware has brought new opportunities to people [3]. Currently, the prosperity of Artificial Intelligence and the Internet of Things (AIoT) brings an amount of demand for chips from AIoT devices [4]. Unlike other devices, such as personal computers and smartphones, AIoT devices in different applications require totally different chips that meet the latency, power, and area needed. Moreover, AIoT devices in different applications have various requirements of latency, power, and area, showing that some devices are more sensitive to power than latency and area, while some devices are more sensitive to area. In other words, the demand for chips from AIoT devices varies and is fragmented.

Therefore, the rapid construction of scene-specific systems-on-chips (SoCs) in the emerging AIoT field is promising [5]. With the development of hardware description languages (HDLs), such as Chisel [6] on Scala, ClaSH [7] on Haskell, and PyRTL [8] on Python, engineers can design hardware in a parameterized and reusable way, which can accelerate the construction of custom SoCs. Moreover, there are a few template-based chip design methods that aim to rapidly construct chips by adjusting the circuit modules and the parameters. FabScalar [9] was developed by Niket K et al. for automatically composing synthesizable register-transfer-level (RTL) designs of arbitrary cores within a canonical superscalar template. The template defines canonical pipeline stages and interfaces among them. A canonical pipeline stage library (CPSL) provides many implementations of each canonical pipeline stage that differ in their superscalar width and depth of subpipelining. An RTL generation tool uses a template and CPSL to automatically generate the overall core of a desired configuration. Rocket Chip [10], proposed by Krste et al., is an open-source SoC design generator that emits synthesizable RTL. It leverages Chisel to compose a library of sophisticated generators for cores, caches, and interconnects into an integrated SoC.

Rocket Chip generates general-purpose processor cores that use the open RISC-V ISA and provides both an in-order core generator (Rocket) and an out-of-order core generator (BOOM). For SoC designers interested in utilizing heterogeneous specialization for added efficiency gains, Rocket Chip supports the integration of custom accelerators in the form of instruction set extensions, coprocessors, or fully independent novel cores. Moreover, Rocket Chip has been taped out (manufactured) eleven times and yielded functional silicon prototypes capable of booting Linux. Sizhuo Zhang et al. presented a framework called composable modular design (CMD) [11] to facilitate the design of out-of-order processors. In CMD, the interface methods of modules provide instantaneous access and perform atomic updates to the state elements inside the module. Modules are composed together by atomic rules that call interface methods of different modules. The atomicity properties of interfaces in CMD ensure composability when selected modules are selectively refined.

The designs of the methods mentioned are compiled into RTL, which can be run on FPGAs or synthesized using standard ASIC design flows. However, existing chip template-based design methods still use synthesis and simulation to obtain the quality of results (QoR) regarding performance, power, and area (PPA) for parameter set design space exploring (DSE), as shown in Figure 1. However, iteration of the parameter set DSE via synthesis and simulation flows is time-consuming. Based on this, this study proposes a rapid and accurate PPA prediction for template-based chip design methods. First, a preprocess for template-based methods obtains the influence of each parameter of the templates on the PPA. Moreover, a PPA prediction model based on multivariate linear regression (ML-PM) is proposed to fit the multiparameter influence on the PPA via a single parameter effect. However, ML-PM does not consider the laws on processor performance. Therefore, a multivariate nonlinear regression prediction model (MNL-PM) based on Amdahl's law is introduced to improve the accuracy of the PPA estimation.

As shown in Figure 2, the proposed method replaces the synthesis and simulation process to reduce the time cost during the parameter design process. An empirical evaluation of the method shows that the PPA prediction for template-based chip design methods can reach 98.60%, 99.19%, and 98.53% accuracy on performance, power, and area, respectively, when compared with a PPA generated via synthesis and simulation flows. To the best of our knowledge, there is no previous work on fast and accurate PPA prediction on HDL design from template-based processor design methods. We believe that the proposed method is a novel and interesting design point in the space of solutions to the agile chip design problem. The main contributions of this work can be summarized as follows:

- The PPA prediction model, ML-PM is proposed to achieve the PPA prediction via the parameter set, in which there are alters for normal multivariate linear regression model.
- The PPA prediction model, MNL-PM based on Amdahl's law is introduced as to improve the accuracy of the PPA estimation.
- The iteration process of the parameters set DSE has been improved with the proposed PPA prediction model, which is time thrifty.

To the best of our knowledge, there is no previous work on fast and accurate PPA prediction on HDL design from template-based processor design methods. We believe that the proposed method is a novel and interesting design point in the space of solution to the agile chip design problem.

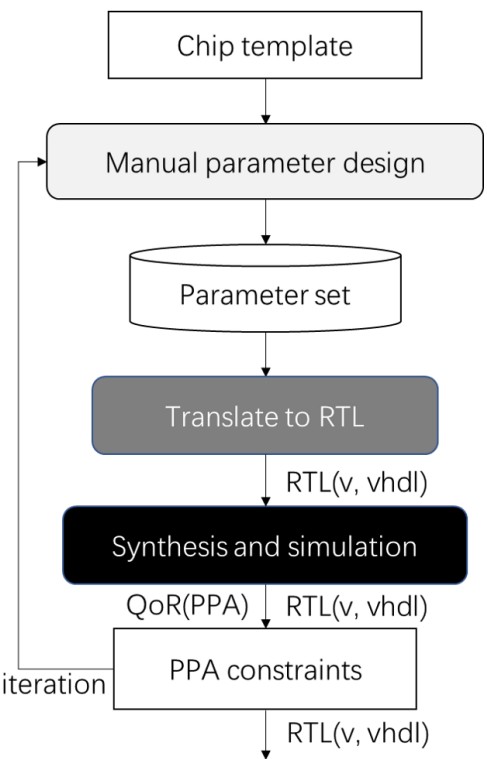

**Figure 1.** The parameter design process.

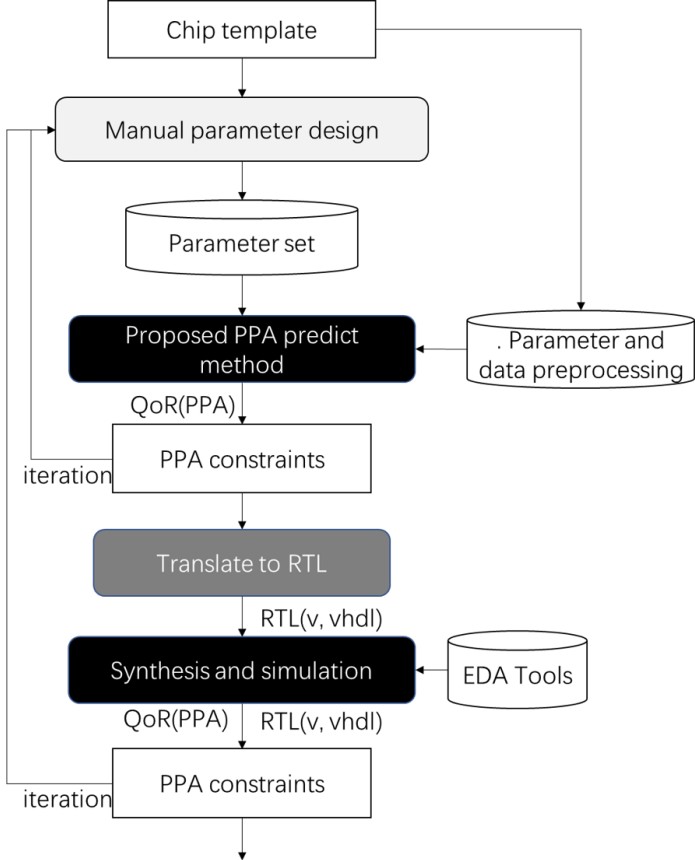

**Figure 2.** The parameter design process with proposed PPA prediction method.

## 2. Related Work

As the AIoT grows, there are demands on processors that have different PPA requirements for various scenarios. Therefore, it is critical to design an AIoT process that meets the PPA needs in the short term. Template-based processor design methods, built on parameterized HDLs, aim to build chip architecture rapidly. However, these methods still require synthesis and simulation flows to obtain the PPA reports and find the proper parameter set, which takes a long time. Based on this, this study proposes a PPA prediction method that leverages the heavily parameterized features of template-based processor design methods to accelerate the iteration of parameter set design.

### 2.1. Parameterized HDLs

Traditional HDLs have a very long learning curve, even for experienced engineers. To solve this problem, new HDL designs focus on reusable and parameterized characteristics. Based on Python, PyRTL's goal is to provide an alternative between these two extremes by enabling programmers to directly specify their hardware designs in a language they already know. MyHDL [12] and PyMTL [13] are also Python-based hardware design tools. MyHDL is built around generators and decoders; the semantics of this embedded language are close to Verilog and allow asynchronous logic and higher-level modeling. Much like traditional HDLs, however, only a structural "convertible subset" of the language can be automatically synthesized into real hardware. Chisel is an elaborate through-execution hardware design language. With support for signed types, named hierarchies of wires, and a well-designed control structure, Chisel is a powerful tool used in some great research projects, including OpenRISC [14]. ClaSH is a hardware description embedded DSL in Haskell. ClaSH also provides an approach suitable for both combinational and synchronous sequential circuits and allows the transformation of these high-level descriptions to low-level synthesizable Verilog HDL.

### 2.2. Template-Based Chip Design Methods

Niket K et al. develop a toolset, called FabScalar, for automatically composing synthesizable RTL designs of arbitrary cores within a canonical superscalar template. The template defines canonical pipeline stages and interfaces among them. A canonical pipeline stage library (CPSL) provides many implementations of each canonical pipeline stage that differ in their superscalar width and depth of subpipelining. Rocket Chip, proposed by Krste A et al., is an open-source System-on-Chip design generator that emits synthesizable RTL. It leverages the Chisel hardware construction language to compose a library of sophisticated generators for cores, caches, and interconnects into an integrated SoC. Rocket Chip generates general-purpose processor cores that use the open RISC-V ISA and provides both an in-order core generator (Rocket) and an out-of-order core generator (BOOM). For SoC designers interested in utilizing heterogeneous specialization for added efficiency gains, Rocket Chip supports the integration of custom accelerators in the form of instruction set extensions, coprocessors, or fully independent novel cores. Sizhuo Zhang et al. present a framework called composable modular design (CMD) to facilitate the design of out-of-order processors. CMD designs can also be compiled into RTL, which can be run on FPGAs or synthesized using standard ASIC design flows. The atomicity properties of interfaces in CMD ensure composability when selected modules are selectively refined. Template-based chip design methods focus on the rapid construction of chips; therefore, modular and parameterized template design is the key point.

### 2.3. PPA Prediction for HLS

The relation between the knobs and the PPA for HLS design is a black box; therefore, machine learning is widely used for PPA prediction in HLS design to accelerate the design process. A. Mahapatra et al. proposed a machine-learning-based simulated annealer method for high-level synthesis design space exploration in 2014 [15]. To foresee the correlation between power consumption and HLS-based applications at an early design

stage, Zhe Lin et al. introduced HL-Pow [16], a power modeling framework for FPGA HLS based on state-of-the-art machine learning techniques. HL-Pow incorporates an automated feature construction flow to efficiently identify and extract features that exert a major influence on power consumption, simply based upon HLS results, and a modeling flow that can build an accurate and generic power model applicable to a variety of designs with HLS. Moreover, W. Rhett Davis et al. presented models for fast and accurate PPA prediction that can reduce the manual optimization iterations with EDA tools in 2021 [17]. They investigated techniques to automate PPA optimization using evolutionary algorithms. For PPA prediction, a baseline model is trained on a known design using Latin hypercube sample runs of an EDA tool, and transfer learning is then used to train the model for an unseen design. With transfer learning, the same accuracy was achieved on a different (unseen) design in only 15 runs, indicating the viability of transfer learning to generalize PPA models. However, PPA prediction for HLS is usually used for prototype design and algorithm verification, while HDLs are preferred for chip tape-out in the industry [1]. On the one hand, HLS approaches promise to raise the level of abstraction and compile regular C/C++ functions into logic elements. There is still no clear way to develop an optimized design without prior hardware engineering experience. Moreover, the parameters for HLS are 3 kinds of knobs, which cannot control circuit details such as the ways cache numbers. On the other hand, PPA prediction methods based on machine learning require large samples for training. Therefore, we propose a rapid and accurate PPA prediction method for HDL designs from template-based chip design methods. On the one hand, there is a describable relation between the parameters of an HDL design and its PPA. The circuit details can be directly controlled by modifying the HDL codes. However, the proposed PPA prediction method requires a few samples to estimate the PPA. Table 1 shows a comparison among the proposed method and the best machine learning model, XGBoost [18], for HLS PPA prediction, which is mentioned in [17].

**Table 1.** The proposed method and the state of art methods for PPA prediction in HLS.

| Method | Accuracy (%) on the PPA | Number of the Samples Needed to Achieve the Method | Parameters for Prediction |
|---|---|---|---|
| XGBoost | 98.23 | 400 | 3 kinds of knobs |
| NLR PPA estimation method | $98.82 = (98.60 + 99.19 + 98.68)/3$ | 42 | Particular parameters in HDLs codes |

### 3. PPA Prediction Method

The proposed method aims to rapidly estimate accurate PPA based on the parameter set of a certain chip design in the early stage of design, which can accelerate the process of proper parameter set design. The method contains two parts: one is the preprocess for a template-based method, and the other is a prediction model for the PPA estimation of the parameter set. In this study, a template-based method, Rocket Chip, is chosen as an example.

#### 3.1. Parameter and Data Preprocessing

As the proposed method aims to predict the PPA of the parameter set for template-based methods, it is critical to determine the correlation among parameters and the design space of the parameter set. In Rocket Chip, for instance, a single core is mainly dependent on the parameters of modules, including the Core, Cache, BTB, and BTB's submodule, BHT. First, only one of the parameters with correlations is selected as their representative. In this way, 27 parameters are selected as the parameter set. Second, to determine the design space of the parameter set, the value range of each parameter is designed, as shown partly in Table 1, as follows.

- For a bool-type parameter, there is only one optional value, and the default value is negated, except for the original value.

- For an int-type parameter, there are two situations. For the first situation, there are two optional values except the original value of itself. Moreover, these three values form a proportional sequence with a common ratio of 2. As in the real design process, the parameters such as the size of memory are changed at a ratio of 2. Moreover, the default values will be the smallest, the middle, or the largest among these three values depending on the analysis of the existing designs in Rocket Chip. For other situations, the default values of these parameters are 0 and are still 0 in the existing designs in Rocket Chip. Then, 1 is set as another optional value of these parameters, similar to the rule for a bool-type parameter.

Moreover, the proposed method requires the individual influence of each parameter on the PPA to calculate the PPA prediction. Therefore, the bold value in Table 2 is designed as the base parameter set, and each parameter is changed one at a time which are set as changed parameter sets to obtain the individual influence on PPA of each parameter. Specifically, the rules to set the values of the base parameters are that for int-type parameters with 3 optional values, the middle values are chosen, and for int-type parameters with value range, 0 and 1 values and bool-type parameters, the set value depends on the existing designs in Rocket Chip. Moreover, Table 3 shows an instance of changed sets partly. In this way, there are only 42 parameter set samples with PPA needed, and the PPA of the parameter set with the design space of $3.918 \times 10^{10}$ is predictable.

**Table 2.** The portion of parameters set with value range.

| Parameter | Module | Value Range |
|---|---|---|
| mulEarlyOut | Core | **true**, false |
| divEarlyOut | Core | **true**, false |
| divSqrt | Core | **true**, false |
| nSets | Icache | 16, **32**, 64 |
| nWays | Icache | 2, **4**, 8 |
| nTLBSuperpages | Icache | 2, **4**, 8 |
| nSets | Dcache | 16, **32**, 64 |
| nWays | Dcache | 2, **4**, 8 |
| nTLBSets | Dcache | 1, **2**, 4 |
| nEntries | BTB | 256, **512**, 1024 |
| updatesOutOfOrder | BTB | true, **false** |
| historyLength | BTB | 1, **2**, 4 |
| counterLength | BTB | 2, **4**, 8 |
| ... | ... | ... |

The bold values are the defaults.

**Table 3.** An instance of the same portion of one changed set.

| Parameter | Module | Value |
|---|---|---|
| mulEarlyOut | Core | **true** |
| divEarlyOut | Core | false |
| divSqrt | Core | **true** |
| nSets | Icache | **32** |
| nWays | Icache | **4** |
| nTLBSuperpages | Icache | **4** |
| nSets | Dcache | **32** |
| nWays | Dcache | **4** |
| nTLBSets | Dcache | **2** |
| nEntries | BTB | **512** |
| updatesOutOfOrder | BTB | **false** |
| historyLength | BTB | **2** |
| counterLength | BTB | **4** |
| ... | ... | ... |

The bold values are the defaults.

### 3.2. Prediction Model Derivation

Having the effect of each parameter of the parameter set on PPA is not enough to predict the PPA of a certain parameter set design. Moreover, the relationship between the multifactor influence and a single-factor effect on the same system is a black box. Therefore, multivariate regression is utilized to fit the multiparameter influence on the PPA via a single parameter effect. The PPA estimate of the parameter set can be calculated via this method and the PPA samples of each parameter. In this study, the fit model is built separately based on a weight model and Amdahl's law.

#### 3.2.1. Multivariate Linear Regression Prediction Model

To build the ML-PM, we assume that each parameter is independent and has three individual weights for performance, power, and area. In detail, for each parameter $p_i$, $i \in (1, n)$, this paper assumes that there are weight $w_i'$, $w_i''$, and $w_i'''$, for performance, power, and area specifically, while $w_i'$, $w_i''$, and $w_i''' > 0$. Taking performance ($P_1$) for instance, $P_1$ is the performance value of the base parameter set. And $P_1$ will turn to $P_{1,m'}$ and fellow equation holding when one parameter $p_m, m \in (1, n)$ is modified to other value in its value range.

$$S = \frac{w_m' \bullet (1 + \beta_m) + \sum_{i=1, i \neq m}^{n} w_i'}{\sum_{i=1}^{n} w_i'} \tag{1}$$

$$P_{1,m}' = P_1 \bullet S \tag{2}$$

In the Equation (1), $\beta_m$ is the influence rate of the modified parameter $p_m$ on $P_1$. And the change rate S can be represented as Equation (3), when multiple parameters $J = \{p_j \ldots\}$, $J \subseteq \{p_1 p_2 \ldots p_n\}$ are modified.

$$S = \frac{\sum_{j, p_j \in J}^{n} w_j' \bullet (1 + \beta_j) + \sum_{i, p_i \notin J}^{n} w_i'}{\sum_{i=1}^{n} w_i'} \tag{3}$$

Although neither the $\beta_i$ nor the $w_i'$ are known, the values of $P_1$ and $P_{1,j}'$ are already known in the former preprocess, therefore, $w_j'$ and $\beta_j$ can be replaced via $\Delta P_{1,j}$ as the Equation (4) below.

$$\Delta P_{1,j} = P_{1,j}' - P_1 = \frac{w_j' \bullet \beta_j}{\sum_{i=1}^{n} w_i'} \bullet P_1 \tag{4}$$

Based on the Equations (3) and (4), the change rate S can be calculated via available information, as the Equation (6) below.

$$S = 1 + \frac{\sum_{j, p_j \in J}^{n} \Delta P_{1,j}}{P_1} \tag{5}$$

And $P_{1,J'}$ can be calculated as Equation (6), when multiple parameters $J = \{p_j \ldots\}$, $J \subseteq \{p_1 p_2 \ldots p_n\}$ are modified.

$$P_{1,J}' = P_1 \bullet S = P_1 + \sum_{j, p_j \in J}^{n} \Delta P_{1,j} \tag{6}$$

As the Equations (7) and (8) show separately, the power ($P_2$) and the area ($A_1$) can be calculated in the same way.

$$P_{2,J}' = P_2 + \sum_{j, p_j \in J}^{n} \Delta P_{2,j} \tag{7}$$

$$A_{1,J}' = A_1 + \sum_{j, p_j \in J}^{n} \Delta A_{1,j} \tag{8}$$

3.2.2. Multivariate Nonlinear Regression Prediction Model

Inspired via Amdahl's law [19], the MNL-PM is proposed to estimate the PPA. Amdahl's law provides the maximum theoretical speedup achievable by a system which reflects the influence of each part in one system to the whole system at the same time. In detail, for each parameter $p_i$, $i \in (1, n)$, the proposed method assumes that there are rates $w_i'$, $w_i''$, and $w_i'''$, each for its proportion in performance, power, and area, while $w_i'$, $w_i''$, and $w_i''' > 0$ and $\sum_{i \in (1,n)} w_i'$, $\sum_{i \in (1,n)} w_i''$, $\sum_{i \in (1,n)} w_i''' < 1$. Taking performance ($P_1$) from PPA for instance, $P_1$, the value of the baseline's performance, will turn to $P_{1,m'}$ and fellow equations holding when parameter $p_m$, $m \in (1, n)$ is modified to other value in its value range.

$$S_{1,m} = \frac{1}{(1 - w_m') + w_m'/(1 + \beta_m)} \tag{9}$$

$$P_{1,m}' = P_1 \bullet S_{1,m} \tag{10}$$

In the Equation (9), $S_{1,m}$ is the change rate and $\beta_m$ is the change rate of the modified $p_m$. And the change rate $S_{1,J}$S can be represented as Equation (11), when multiple parameters $J = \{p_j \dots\}$, $J \subseteq \{p_1 p_2 \dots p_n\}$ are modified.

$$S_{1,J} = \frac{1}{\left(1 - \sum_{j,p_j \in J}^n w_j'\right) + \sum_{j,p_j \in J}^n \frac{w_j'}{1 + \beta_j}} \tag{11}$$

Although neither the $\beta_i$ nor the $w_i'$ are known, the values of $P_1$ and $P_{1,j}'$ are already known in the former preprocess. And $S_{1,j}$ can directly expressed via $P_1$ and $P_{1,j}'$, as the Equation (14) below.

$$S_{1,j} = P_{1,j}'/P_1 \tag{12}$$

Based on the Equations (11) and (12), the change rate S can be calculated via available information, as the Equation (13) below.

$$S_{1,J} = \frac{1}{\sum_{j,p_j \in J}^n \frac{P_1}{P_{1,j}'} - k + 1} \tag{13}$$

And $P_{1,J'}$S can be calculated as Equation (14), when multiple parameters $J = \{p_j \dots\}$, $J \subseteq \{p_1 p_2 \dots p_n\}$ are modified.

$$P_{1,J}' = P_1 \bullet S_{1,J} = P_1 \bullet \frac{1}{\sum_{j \in J} \frac{P_1}{P_{1,j}'} - k + 1} \tag{14}$$

As the Equations (15) and (16) show separately, the power ($P_2$) and the area ($A_1$) can be calculated in the same way

$$P_{2,J}' = P_2 \bullet \frac{1}{\sum_{j \in J} \frac{P_2}{P_{2,j}'} - k + 1} \tag{15}$$

$$A_{1,J}' = A_1 \bullet \frac{1}{\sum_{j \in J} \frac{A_1}{A_{1,j}'} - k + 1} \tag{16}$$

**4. Experimental Results**

*4.1. Implementation Details*

In this part, experiments are designed to test the performance of the proposed PPA prediction method for template-based processor design method. For the PPA prediction method there are parameter set samples designed for the NLR PPA prediction method testing. And the QoRs of these samples, obtained via the EDA tools, are seemed as the

standard value, while the QoRs of these samples, obtained via the proposed proposed method, are seemed as the prediction value.

The template-based method, Rocket-Chip Generator is chosen as a case. To obtain the standard PPA value of the samples, the synthesis is based on the EDA tool with the 28 nm technology and other constraints are defined in the tcl file; and simulation with benchmark program, specifically, the benchmark program is the Dhrystone offered via the Rocket-Chip. In this way, the standard PPA values of the 42 sample during the preprocess are obtained.

### 4.2. PPA Prediction for the Parameter Set

First, the samples are evenly sampled from the design space of the parameter set. Specifically, the number of parameters that are chosen to change ranges from 2 to 27, and for each case, there are 10 random samples. Therefore, there are 26 × 10 = 260 samples in total. Then, the ML-PM and MNL-PM are both used to estimate the PPA of these 260 samples. Moreover, the standard PPA value is also generated via synthesis and simulation flows, and Table 4 shows the PPA prediction performance for the parameter set of both the ML-PM and MNL-PM. Moreover, the accuracy is calculated via (17), in which the prediction is the PPA value obtained via the proposed NLR method andthe standard is the PPA value obtained via the EDA tools.

$$Accuracy(\%) = (1 - \frac{|prediction - standard|}{standard}) \times 100\% \qquad (17)$$

**Table 4.** The accuracy of the PPA prediction method.

| Prediction Model | Accuracy of Performance (%) | Accuracy of Power (%) | Accuracy of Area (%) |
|---|---|---|---|
| ML-PM | 98.42 ± 2.20 | 99.15 ± 0.97 | 98.53 ± 1.49 |
| MNL-PM | 98.60 ± 2.17 | 99.19 ± 0.83 | 98.68 ± 1.31 |

The results show that MNL-PM performs better on the PPA estimation than ML-PM with higher accuracy and smaller standard deviation. Furthermore, the performance on the PPA prediction with the different number of changed parameters of ML-PM and MNL-PM is shown in Figure 3. The results shows that both proposed method works better when the number of changed parameter is less than 11, and the proposed method works mostly stable on the power prediction. As the results of Equations (1) and (11) are similar when $\beta_m$ is small, the main reason for this fact is that the small value range of each parameter in the parameter set design space, which causes the influence of each parameter is little.

PPA Prediction for Parameters in the Same Module

In a real processor design process, the changes to an existing design in one time iteration usually appear in the same module. The samples of the parameter set design space in the experiment above cannot represent this behavior well. Therefore, the samples of the changed parameters in the same module are sampled in this section. In this experiment,there are 11, 5, 7, and 4 parameters in the Core, ICache, Dcache, and BTB modules, respectively. Taking the Core module as an example, the number of parameters that are chosen to change ranges from 2 to 11, and for each case, there are 4 random samples. In the same way, the samples for the other three modules are generated, and there are 92 samples. Moreover, the standard PPA value is also generated of these 92 samples in total. Then, the ML-PM and MNL-PM are both used to estimate the PPA via synthesis and simulation flows, and a scatter plot of the PPA prediction results for the parameter in the same module are shown in Figure 4.

In Figure 5, MNL-PM performs better when the changed parameters are in the same module than randomly distributed.The results shows that there is irregular influence on the ML-PM.

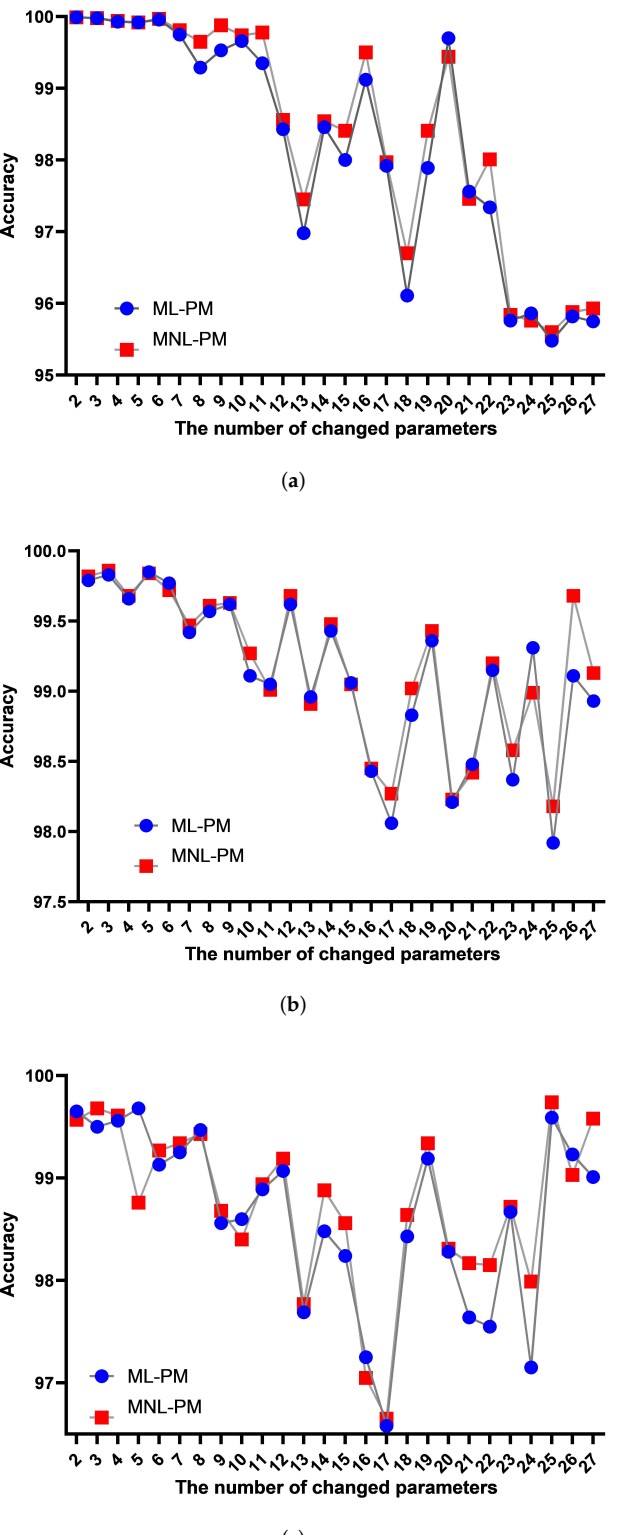

**Figure 3.** The performance on the PPA prediction with the different number of changed parameters. ((**a**) for performance, (**b**) for power, and, (**c**) for area).

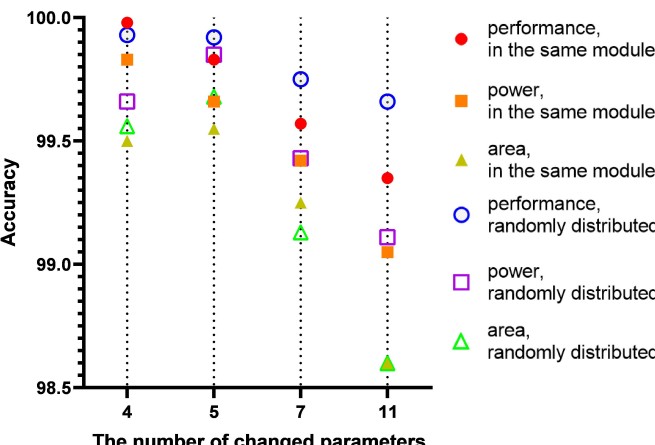

**Figure 4.** ML-PM on the same number of changed parameters in the same modules and randomly distributed.

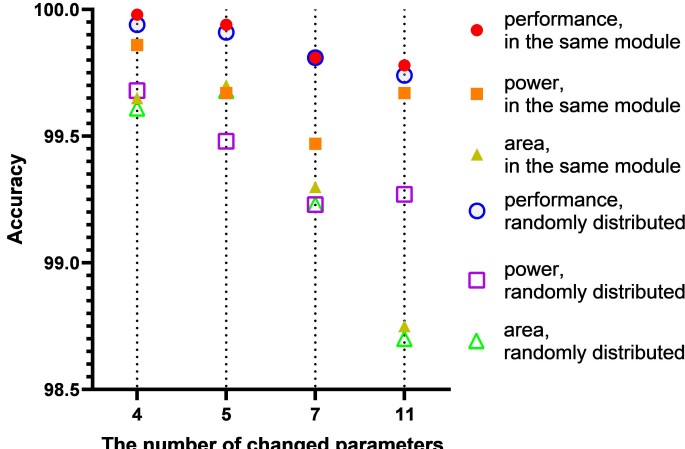

**Figure 5.** MNL-PM on the same number of changed parameters in the same modules and randomly distributed.

## 5. Discussion

To meet the demand of processors with different PPA requirements for various scenarios in the AIoT, this paper presents a rapid and accurate PPA prediction method for template-based processor design methods. The two prediction models, the ML-PM and MNL-PM, are built to estimate the PPA for certain parameter designs. The experimental results show that the MNL-PM is better than the ML-PM and both prediction models work well in the instance of the single-core chip template in the Rocket Chip Generator. However, as limitation of the value range is small and the limitation of the single-core chip template whose PPA range is small, the advantage of the MNL-PM is not obvious as shown in Table 4. Therefore, in our further study, we will work on improve the chip template and enlarge the value range of the parameters in the template.

## 6. Conclusions

To meet the demand of processors with different PPA requirements for various scenarios in the AIoT, this paper presents a rapid and accurate PPA prediction method for template-based processor design methods. In this method, there is the preprocess for the template-based methods at first. Then, two prediction models, the ML-PM and MNL-PM, are built to estimate the PPA for certain parameter designs. An empirical evaluation of the method shows that the PPA prediction for the template-based chip design methods can

reach 98.60%, 99.19%, and 98.68% accuracy on performance, power, and area, respectively, when compared with PPA generated via synthesis and simulation flows. In future studies, on the one hand, the value range of the parameters in the parameter set will be expanded, and on the other hand, an exploration algorithm will be introduced to replace the manual parameter design in the parameter design process, as shown in Figure 2.

**Author Contributions:** Conceptualization, M.T., L.H. and W.C.; methodology, M.T., L.H. and W.C.; software, M.T.; validation, M.T., L.H. and W.C.; formal analysis, M.T., L.H. and W.C.; investigation, M.T., L.H. and W.C.; resources, M.T., L.H. and W.C.; data curation, M.T., L.H. and W.C.; writing— original draft preparation, M.T.; writing—review and editing, M.T., L.H. and W.C.; visualization, W.C.; supervision, M.T., L.H. and W.C.; project administration, M.T., L.H. and W.C.; funding acquisition, L.H. and W.C. All authors have read and agreed to the published version of the manuscript.

**Funding:** This research was funded by he Independent and open subject fund (grant no. 202101-10) from State Key Laboratory of High Performance Computing, National Nature Science Foundation of China (NSFC) under Grant No. 62090023 and No. 618722374, and the National Key Research and Development Program of China (No. 2018YFB0204301).

**Institutional Review Board Statement:** Not applicable.

**Informed Consent Statement:** Not applicable.

**Data Availability Statement:** Not applicable.

**Acknowledgments:** This work was supported by the Independent and open subject fund (grant no. 202101-10) from State Key Laboratory of High Performance Computing, National Nature Science Foundation of China (NSFC) under Grant No. 62090023 and No. 618722374, and the National Key Research and Development Program of China (No. 2018YFB0204301).

**Conflicts of Interest:** The authors declare no conflict of interest.

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
