# Peer review of "Rapid and Accurate PPA Prediction for the Template-Based Processor Design Methods"

_applsci, doi:10.3390/app12168383_

Round 1

Reviewer 1 Report

As an alternative to XGboost, it would be good to see how methods such as random forest, adaptive boosting, kNN and Multi Layer Perceptron yield results. It would be good to share the data for verification.

In the results section, authors only share findings of the proposed method. It would be good to see the results of other alternative/competitor methods.
For the algorithmic papers, presenting a benchmarking table is a kind of tradition. 

Author Response

As an alternative to XGboost, it would be good to see how methods such as random forest, adaptive boosting, kNN and Multi Layer Perceptron yield results. It would be good to share the data for verification.

Response:

Thank you for the comments. However, the work of  XGboost is for the HLS method and our work is for the HDL design. Thus, how methods such as random forest, adaptive boosting, kNN and Multi Layer Perceptron yield results in HLS will not be discussed in this paper.

In the results section, authors only share findings of the proposed method. It would be good to see the results of other alternative/competitor methods.

Response:

Thank you for the comments. However, to the best of our knowledge, there is no previous work on fast and accurate PPA prediction on HDL design from template-based processor design methods.  Thus, we compared our work with the prediction methods for HLS which are related. 

For the algorithmic papers, presenting a benchmarking table is a kind of tradition. 

Response:

Thank you for the comments. However, to the best of our knowledge, there is no previous work on fast and accurate PPA prediction on HDL design from template-based processor design methods. Therefore, we proposed ML-PM and MNL-PM for comparison.

Reviewer 2 Report

This is a very interesting and good piece of work. However, it should be refined to be of maximum use for the readers.

First of all, your approach is reasonable since multivariate regression, either linear or nonlinear, is a method of supervised machine learning. It was already shown in the literature that machine learning works for the problems discussed in your paper.

Second, scientific findings should be  reproducible. Therefore, you should provide more information about your approach, eg. what kind of nonlinear regression models did you use. Additionally, you  could provide regression matrices upon request to judge, whether the models are statistically valid. 

Remarks concerning mainly typos:

Line 61                PM-MLR: explain abbreviation

Line 69                “there is no previous work on fast and accurate PPA prediction on HDL design”

please, discuss this statement with respect to the following link

https://www.synopsys.com/content/dam/synopsys/implementation&signoff/datasheets/rtl-architect-ds.pdf

Lines 68-71        “To the best of our knowledge, there is no previous work on fast and accurate PPA prediction on HDL design from template-based processor design methods. We believe that the proposed method is a novel and interesting design point in the space of solutions to the agile chip design problem.”

Lines 80-83        “To the best of our knowledge, there is no previous work on fast and accurate PPA prediction on HDL design from template-based processor design methods. We believe that the proposed method is a novel and interesting design point in the space of solution to the agile chip design problem.”

Line 206+1                       we assumes->assume

Line after eq(3)               is known->are known, also after eq(11) and so on

Line before eq(4)           can be replace->replaced

Line after 207                  NL-MRAM, please explain
                                          Amdahl's law [? ]

Line 210                           experiments are design->designed

Line 218                           and other constraints in defined in the tcl file;-> are defined

Line 247                           there are 92 f these->of

Line 248                           generated vsamples->?

Line 249                           oia synthesis->? Via

Please, check your equations: the circle operator seems to be a simple multiplication symbol.
The “square” on the left-hand side of eq (4) can obviously be omitted.

Author Response

please, discuss this statement with respect to the following link

Response:

Thank you for the comments. First of all, our work deals with HDL  design from template-based processor design methods, such as Rocket Chip, while the RTL-Architect from this link deals with RTL. Moreover, the computational complexity of our work is O(1), thus, our work can still speed up the design process of the parameters in HDL  design from template-based processor design methods when  RTL-Architect is applied.

Line 61                PM-MLR: explain abbreviation

Line 206+1                       we assumes->assume

Line after eq(3)               is known->are known, also after eq(11) and so on

Line before eq(4)           can be replace->replaced

Line after 207                  NL-MRAM, please explain
                                          Amdahl's law [? ]

Line 210                           experiments are design->designed

Line 218                           and other constraints in defined in the tcl file;-> are defined

Line 247                           there are 92 f these->of

Line 248                           generated vsamples->?

Line 249                           oia synthesis->? Via

Please, check your equations: the circle operator seems to be a simple multiplication symbol.
The “square” on the left-hand side of eq (4) can obviously be omitted.

Response:

Thank you for the comments. We modify all the problems above.